# Fluorescence-Guided Surgery in Pediatric Oncology: Current Practice and Future Directions

**DOI:** 10.3390/cancers18010149

**Published:** 2025-12-31

**Authors:** Dominique C. Simons, Lorenz H. M. van Schalkwijk, Michiel A. J. van de Sande, Alexander L. Vahrmeijer, Marc H. W. A. Wijnen, Alida F. W. van der Steeg, Willemieke S. F. J. Tummers

**Affiliations:** 1Princess Maxima Center for Pediatric Oncology, 3584 CS Utrecht, The Netherlandsa.f.w.vandersteeg@prinsesmaximacentrum.nl (A.F.W.v.d.S.); 2Department of Orthopedic Surgery, Leiden University Medical Center, 2333 ZA Leiden, The Netherlands; 3Department of Surgery, Leiden University Medical Center, 2333 ZA Leiden, The Netherlands

**Keywords:** fluorescence imaging, fluorescence-guided surgery, tumor-specific, indocyanine green, pediatric, surgical oncology

## Abstract

Surgical removal of tumors in children is often difficult because their tumors can grow into nearby healthy tissues and lie close to vital structures. Surgeons must balance removing as much tumor as possible while preserving healthy anatomy. Fluorescence-guided surgery is an emerging technique that uses (non-)specific imaging agents and imaging systems to visualize tumors, blood flow, and vital structures during an operation, potentially improving surgical precision. Although this approach has been widely studied in adults, its use in pediatric oncology is still limited. This review outlines current evidence on how fluorescence guidance is being applied in children, including its benefits and limitations, and discusses the unique challenges of developing more specific imaging agents for pediatric tumors. It also explores strategies to advance this technology toward routine clinical use. If these efforts are successful, fluorescence-guided surgery may enhance surgical accuracy, reduce recurrence risk, and improve long-term outcomes for children with cancer.

## 1. Introduction

For pediatric oncology surgeons adhering to the fundamental principles of surgical oncology—which involve radical resections, accurate disease staging through detection of lymph nodes and avoiding injury to critical structures—can be extremely challenging [1,2]. The challenges mainly lie in infiltrative tumor growth, the size of the pediatric patient and the proximity of critical anatomical structures.

As a consequence, the success of these surgical resections largely depends on the surgeon’s expertise, guided by visual and tactile feedback during the procedure [3]. Irradical resections may lead to incomplete tumor removal, thereby increasing the risk of local recurrence and the need for adjuvant treatment intensification. Moreover, surgical imprecision can result in unnecessary damage, for example, to vital structures and severe postoperative functional impairment. Apart from increasing local recurrence-free and overall survival rates, complete resections may help reduce total dosages of adjuvant chemo- and or radiotherapy. This is particularly important in pediatric patients, who are especially vulnerable to long-term adverse effects of cancer treatment, such as impaired growth and development, organ dysfunction, and the development of secondary malignancies [4].

The inherent challenge of balancing the extent of resection—avoiding both undertreatment and overtreatment—underscores the critical need for intraoperative technologies that enhance surgical precision and support real-time clinical decision making. In this context, both for the adult and pediatric population, there is a growing interest in intraoperative visualization techniques, such as fluorescence-guided surgery (FGS). FGS enables real-time visualization of vital anatomical structures or specific tissues of interest. This technique is based on the Stokes shift principle [5], whereby an external light source excites a fluorescent molecule to a higher energy state. As the molecule returns to its ground state, it emits a photon of lower energy. In clinical practice, after intravenous or local administration of a fluorescent agent, the emitted light is captured using a near-infrared (NIR) imaging system and displayed on the screen in real time (Figure 1).

Fluorescent agents used in FGS can be divided into two main categories: non-targeted and targeted agents. Commonly used non-targeted agents include the FDA- and EMA-approved indocyanine green (ICG), methylene blue (MB) and sodium fluorescein. These agents have been increasingly implemented in adult and pediatric surgical oncology for various applications, including lymph node identification, perfusion assessment, and solid tumor delineation [6]. To improve the precision of FGS and to explore broader clinical indications, research over the last years has focused on the development of targeted fluorescent agents, including antibodies, antibody fragments, peptides, or small molecules conjugated to a fluorescent dye. These agents are designed to bind selectively to tissue-specific biomarkers, thereby reflecting distinct cellular processes. Adequate target validation is particularly relevant in a pediatric context, as pediatric tumors frequently have an embryonal origin and differ substantially from adult malignancies in their biological characteristics. This target validation requires linking fluorescence to histology to confirm binding specificity and to assess receptor heterogeneity during image processing. Furthermore, applying uniform definitions for quantitative metrics is necessary to account for the unique biological variability encountered intraoperatively. Ultimately, targeted fluorescence imaging aims to enhance the accuracy of tissue delineation and improve surgical outcomes.

While in adult oncologic surgery, numerous clinical trials are investigating the use of targeted fluorescent agents, the application of targeted agents for FGS in pediatric oncology remains in its infancy [7]. Most clinical studies have concentrated on non-targeted agents, primarily evaluating safety, proof-of-concept, and the technical or diagnostic performance of this technique. However, evidence remains limited regarding understanding the relation between pathology and fluorescence of either non-targeted or targeted agents, its impact on intraoperative decision making and the added value for the patient.

In this review, we provide a comprehensive and up-to-date overview of the current literature on FGS in pediatric oncology, organized by clinical indication and highlighting recent technological and translational advancements in the field (Figure 2). Unique in this review, we specifically focus on the unique challenges inherent to pediatric oncology. Finally, we propose concrete strategies to accelerate clinical translation and critically assess the potential clinical value of FGS in pediatric oncologic surgery.

## 2. Materials and Methods

A literature search was conducted in Pubmed and Embase for articles in English published up to August 2025 that reported the use of fluorescence imaging in pediatric oncology. The search strategy included the terms ‘Fluorescence’, ‘Pediatrics’, ‘Neoplasms’, and ‘Surgery’, and was supplemented with relevant title and abstract keywords to provide a comprehensive review of relevant articles. The search string was developed with our institutional medical library and is provided in the Appendix A. Our narrative review excluded: studies that were not related to (surgical) oncology, adult population, non-English language, case reports or reviews, brain or eye neoplasms, or publications unrelated to (fluorescence) image-guided surgery. The study selection involved two independent researchers (D.S., L.S.) who identified all relevant articles. Conflicting interpretations were resolved by consulting a third reviewer (W.T.) to reach a consensus. Of the 1046 identified articles in our initial search, 40 full-text publications were included based on the search string. Based on the screened articles, studies were grouped into three intended clinical applications: lymph node identification, vital structure imaging, or tumor tissue visualization. Afterwards, the studies were divided into non-targeted or targeted agents (Table 1). Quality and risk of bias assessment was not performed, as this narrative review aimed to describe current practice and propose strategies for clinical translation.

## 3. Identification of Lymph Nodes

Fluorescent dyes have been used to facilitate lymphatic mapping and sentinel lymph node biopsy in surgical oncology for decades. Until recently, the sentinel lymph node procedure (SNP) in pediatric oncology involved the use of a radiotracer and an optional blue dye injection around the tumor site, commonly MB, for visual guidance. MB is visible without imaging system; however, an important downside of MB is the fact that handling the injection site can lead to diffuse blue staining of the surroundings, obscuring the target area. Moreover, MB is associated with a relatively low efficacy (60%), limited penetration depth, and risk of severe allergic reactions [48,49,50]. Therefore, ICG guidance has gained increasing interest as a safer and more effective alternative to blue dyes, as reported by Campwala et al. and Johnston et al. [8,9]. Thus far, the use of ICG during SNP in children has been described for melanoma, squamous cell carcinoma, and soft tissue sarcomas located in the extremities, head and neck, or trunk.

Jeremiasse et al. reported that ICG was particularly useful for nodes in the inguinal area, head and neck regions, and in transit sites compared to the axilla, likely due to its maximum penetration depth leading to reduced utility for deeper-lying sentinel lymph nodes [10]. To overcome this limited penetration depth, novel optical depth-sensing approaches, such as dual-band systems capable of simultaneously detecting ICG and PpIX, are being developed to enhance deep-tissue fluorescence [51]. Moreover, frequently cited additive values included the detection of additional (sentinel) lymph nodes, and the transcutaneous visualization of sentinel lymph nodes when the agent was administered prior to incision [11], which occurred in about two-thirds of cases. This capability may aid in surgical incision planning (Figure 3) and potentially reduce time under anesthesia. In most studies, ICG was combined with Technetium-99 m nanocolloid to enhance detection of deeper lymph nodes. In adults with breast cancer, ICG alone achieved comparable sentinel lymph node detection [52], which may be relevant for future pediatric applications.

In pediatric oncology, sampling multiple lymph nodes is a key component of staging certain pediatric tumors, primarily for Wilms tumors and paratesticular rhabdomyosarcomas [53,54,55,56], which may influence intensification of treatment. Thus far, lymph node harvesting for pediatric testicular rhabdomyosarcoma under ICG guidance has only been reported during retroperitoneoscopic surgery, as described by Mansfield et al. and Pio et al. [12,13], where ICG was injected intraoperatively into the spermatic cord under ultrasound guidance to enhance visualization of lymphatic drainage pathways. The first node showing ICG avidity was designated as the sentinel node and was tumor-positive in some cases. Similarly, ICG-guided lymphatic mapping and nodal sampling have been described in Wilms tumor [14,15], achieved through peri-hilar injection or ipsilateral intra-parenchymal injection. These studies showed that the use of ICG in fluorescence-guided surgery increases the number of nodes detected. This is an important consideration given current guidelines recommending sampling of ≥7 nodes in Wilms tumor, despite a median of only 4 typically being reported [54,57]. Although the intra-parenchymal approach was technically more successful than the peri-hilar approach, it can be more challenging in tumors lacking clearly visible healthy renal parenchyma, for example, during upfront resection of larger tumors.

In general, ICG fluorescence in pediatric oncology enables the visualization of lymph nodes for mapping purposes. In this context, FGS is used to identify anatomical structures rather than malignancy, making ICG a safe and valuable technique for surgeons. However, whether it actually improves the precision of nodal sampling or reduces the risk of treatment-related morbidity and complications such as vascular injury is currently being investigated in an ongoing randomized controlled trial (ISRCTN26150156).

## 4. Imaging of Vital Structures

### 4.1. Non-Specific Tissue Imaging

Within pediatric oncology, identification of vital structures in proximity of the tumor, such as blood vessels, nerves and ureters is essential for safe tumor excision. Fluorescence can also play a critical role in perfusion imaging (Figure 4). Aung et al. evaluated the use of ICG during rotationplasty to visualize perfusion during pediatric sarcoma resection in three patients [16]. Vascular management is essential for a successful rotationplasty and prevents tissue necrosis and complications. ICG allows surgeons to visualize blood vessels, including the perivascular system of nerves, in real time during tumor resection and rotationplasty, providing insight into tissue perfusion and helping to reduce postoperative complications such as wound healing problems or flap necrosis. During surgery, it helps to identify anatomical variations. Another advantage is the possibility of repeated administration, which can be useful when dissecting neurovascular bundles from the tumor. The authors concluded that ICG fluorescence serves as a safe adjunct that supports surgical decision making by providing essential information on tissue perfusion when planning complex ablative and reconstructive procedures, such as rotationplasties.

To aid intraoperative guidance during pediatric minimally invasive procedures, Esposito and colleagues have consistently highlighted the value of ICG fluorescence imaging. Across their series, ICG was applied intravenously during the resection of abdominal masses and lymphomas. They concluded that ICG was useful for identification of vascular anatomy of the abdominal mass, for determining the plane of resection during mesenteric division, and for perfusion assessment of the bowel [17]. Esposito and colleagues found that ICG helped to assess the vascularity of the ovary, salpinx and uterus after tumor excision [18,19,20]. For paratubal lesions, ICG helped to check the vascular permeability of the fallopian tube after resection [21]. Intraoperative ICG injection, allowed for fast (~60 s) and safe visualization of both critical anatomical structures and pathological entities, thereby potentially preserving organ function and patient’s safety.

Collectively, these studies demonstrate that ICG fluorescence imaging could serve as an adjunct in pediatric oncologic surgery by enabling real-time visualization of vascular anatomy and soft tissue perfusion during tumor resections and reconstructive procedures. Currently, this technique cannot replace image-guided localization techniques; however, ICG could assist surgeons in real-time intraoperative decision making in complex pediatric oncological surgery, allowing them to perform precise resections with the goal of mitigating complications such as ischemia, wound healing problems, or inadvertent injury to adjacent structures. Its rapid onset, possibility for repeated dosing, and ability to distinguish normal from pathological structures support precise resections, reduce complications, and facilitate organ preservation, thereby enhancing both surgical safety and outcomes in complex procedures.

### 4.2. Advances in Tissue-Specific Imaging Approaches

While no studies in pediatric oncology have yet reported the use of tissue-specific fluorescence to identify vital anatomical structures, several promising advances have emerged in adult surgery [58]. Firstly, multiple approaches for accurate intraoperative identification of nerve tissue to avoid iatrogenic nerve injuries and corresponding morbidities have been made [59]. Lee et al. explored the use of bevonescein (ALM-488), a small peptide labeled with fluorescein binding to the structural extracellular matrix component of the nerves, in head and neck surgery in a phase 1 trial (NCT04420689) [60]. Specific binding to nerve tissue was observed and the injection of bevonescein was considered safe. Consequently, bevonescein is currently being explored in phase 2 clinical trials for nerve visualization in head and neck surgery (NCT06227585, including adults and children aged 16 years and older) and for abdominopelvic surgery in adults (NCT06662097). In addition, nerve-specific oxazine based fluorophores are being explored in pre-clinical studies [61]. Imaging the nerves can be beneficial for different surgical indications, making these studies highly relevant.

Secondly, intraoperative identification of the ureters is gaining interest, for example, to prevent injury during laparoscopic abdominopelvic procedures in adults. Instead of using ureteral stents, which can itself result in complications such as infection, prolonged need for antibiotics or iatrogenic ureteral damage [62,63], NIR fluorescence imaging could provide non-invasive and real-time visualization of the ureters during surgery. De Valk et al. addressed this challenge with a novel zwitterionic NIR fluorophore, nizaracianine triflutate or ZW800-1, specifically engineered for ureter visualization [64]. Unlike conventional highly anionic NIR dyes, ZW800-1 facilitates exclusive renal clearance, thereby enabling clear and consistent ureter delineation for several hours in real time. Currently, a phase 2/3 trial (NCT06101745) is running to investigate the safety and effectiveness of ZW800-1 in ureter visualization. Moreover, Farnam et al. developed IS-001, a cyanine NIR ureter probe for intravenous injection [65]. In this phase 1 study, they concluded that IS-001 was safe and renally excreted, allowing enhanced ureter visualization. A phase 3 study is currently evaluating the safety and efficacy of IS-001 for ureter delineation in robotic-assisted gynecological surgery (NCT05954767). Both ZW800-1 and IS-001 enable clear and consistent real-time ureter delineation using clinically available imaging platforms. However, within pediatric surgical oncology, ZW800-1 is of particular interest given its efficacy at low doses (1–2.5 mg ZW800-1 compared to 10–40 mg IS-001) and its prolonged imaging window (3 h with 2.5 mg ZW800-1 compared to 1 h with 30 mg IS-001). These properties not only minimize exposure risk but also provide greater flexibility during complex and lengthy procedures, making it especially well-suited for pediatric applications.

As discussed above, tissue-specific agents hold significant clinical potential; however, their implementation in pediatric practice has not yet been realized.

## 5. Visualization of Pediatric Tumors

### 5.1. Non-Specific Imaging

Another indication of interest for fluorescence imaging is the identification of the tumor location and tumor margins during surgery, both at the primary site and metastatic sites. Clinically approved fluorescent agents, such as ICG and sodium fluorescein, have been employed for this purpose in surgical oncology. Hereafter, the indications are described per tumor type.

#### 5.1.1. Hepatoblastoma

NIR imaging with ICG is increasingly utilized in pediatric hepatoblastoma surgery, where it serves as a complementary tool for identifying multifocal lesions and the demarcation of tumor tissue. ICG shows prolonged retention in tumor tissue compared to hepatocytes due to leaky vascular capillaries and defective lymphatic clearance (i.e., the enhanced permeability and retention (EPR) effect) [66], which hinders its excretion into bile. In most studies, ICG was intravenously administered at 24–96 h before surgery to minimize background fluorescence, with dosages ranging from 0.1–0.5 mg/kg [22,23,24,25,26,27,28,29]. ICG-guided surgery enabled sensitive detection of primary and residual hepatoblatoma, liver satellite lesions, and metastases including millimeter-sized nodules in the abdominal cavity and lungs [30,31,32,33,67]. In addition, ICG fluorescence helped surgeons in visualizing lesions in the liver that were not identified on preoperative imaging. Although ICG fluorescence is highly sensitive, false positivity may lead to unnecessary resection and increased site-specific morbidity, such as hemorrhage and biliary complications in the liver [68,69,70]. Moreover, pretreated lesions often showed heterogeneous or absent fluorescence but based on limited research no clear association with pathology has been seen thus far.

#### 5.1.2. Renal Tumors

ICG-guided nephron-sparing surgery (NSS) is feasible for pediatric renal cancers, with ICG administered intravenously either the day before surgery (1.5 mg/kg) or a fixed intraoperative intravenous dosage (2.5 or 5 mg) [34,35]. In both case series, Wilms tumor and renal cell carcinoma were hypofluorescent and demonstrated an ‘inverse pattern’ of NIR signal. This was attributed to the downregulation of bilitranslocase, which is highly expressed in normal renal tubules and mediates uptake of organic anions, including ICG [71,72]. In contrast, ICG fluorescence was higher in a malignant rhabdoid tumor of the kidney relative to its healthy parenchyma, likely as a result of the EPR effect [66]. NIR-guided tumor localization during NSS succeeded in most kidneys and could potentially reduce time under anesthesia; however, it has not yet shown to improve the rate of margin positivity, which can lead to an escalation of therapy (e.g., adjuvant radiotherapy) to mitigate the associated risk of local recurrence. Additionally, conditions such as tumor adhesions to perirenal tissues and reduced blood supply due to renal artery embolization may influence fluorescence patterns, making this technique less suitable in some cases. Most Wilms tumor cases received upfront chemotherapy, but histological responses varied widely, making it unclear how practical NIR imaging with ICG is for delineating the tumor–renal boundary after neoadjuvant treatment, which is a challenge for all solid tumors [73]. Furthermore, the ICG avidity of pulmonary Wilms tumor metastases has been inconsistent as described in a few patients [34,36].

#### 5.1.3. Abdominal Tumors and Lymphoma

Esposito et al. recently described the use of ICG during laparoscopic excision of abdominal masses in pediatric patients [17]. In this study, ICG was administered intravenously to visualize the vascular anatomy of the abdominal mass during primary tumor resection. Additionally, they found that ICG helped to identify adnexal tumors, which appeared hypofluorescent relative to healthy tissue [18,19,20]. This technique helped the surgeon define and preserve the resection margins, guiding surgical decision making for potential ovarian-sparing surgery. Intraoperative ICG injection, allowed for fast (~60 s) visualization of both critical anatomical structures and pathological entities, thereby potentially positively impacting oncological radicality while preserving organ function and patient’s safety.

#### 5.1.4. Bone and Soft Tissue Sarcoma

Primary bone and soft tissue sarcomas can be visualized using NIR imaging in adults, with a handful of adolescents included. The application of ICG was regarded useful for achieving complete curettage of locally aggressive benign bone tumors [37], such as giant cell tumor of bone. It was considered particularly important for intralesional curettage, as this has been associated with higher rates of local recurrence compared to wide resection [74], but carries a lower risk of complications. Additionally, the use of fluorescence also frequently helped surgeons identify tumor residue in the wound. ICG uptake was observed in a substantial proportion of malignant bone tumors in a case series [38], which has been suggested to reduce the need for extensive bone resection. ICG fluorescence imaging also guided skull bone tumor resection by visualizing occult tumors, enabling complete removal with appropriate margins, and thus, holding promise for other anatomical sites [39]. In multiple centers, ICG has been used for the identification of metastases of high-grade pediatric sarcomas [36,40,41,67], with intravenous injection (0.5–1.5 mg/kg) frequently administered the day before surgery to reduce background noise. Pulmonary lesions of osteosarcoma, Ewing sarcoma, rhabdomyosarcoma, and non-rhabdomyosarcoma, containing viable disease, fluoresced during surgery (Figure 5). This is promising for procedures with limited or no tactile feedback, such as thoracoscopy, and may help identify additional subpleural nodules missed by preoperative computed tomography (CT) or palpation during open surgery. In some children with osteosarcoma, lesions containing sporadic viable tumor cells either did or did not fluoresce, and lesions without viable tumor did not fluoresce [41]. For now, it is suggested that ICG can be used as an adjunct to other localization methods but cannot replace interventional radiology techniques, such as CT-guided placement of a hook, wire, or injection of EVOH polymer. However, these methods also carry the risk of mispositioning, migration, pneumothorax, and pain [40].

#### 5.1.5. Peripheral Nerve Sheath Tumors

The feasibility of fluorescence-guided resection of peripheral nerve sheath tumors has been reported by Vetrano et al., with an intravenous injection of 1 mg/kg sodium fluorescein (560 nm) after intubation [42]. This case series included young adults with schwannomas and neurofibromas, primarily located in the extremities or the brachial plexus region. In peripheral nerves, fluorescein may extravasate into the endoneurium, similar to tumors in the central nervous system, where the blood–brain barrier is disrupted. In this series, fluorescein enabled delineation and resection of tumor tissue after opening the tumor pseudocapsule, while functionally intact nerve fibers were identified using direct intraoperative electrophysiological stimulation. Neurofibromas were removed piecemeal, and fluorescein was regarded useful for detecting small tumor remnants that were not visible using ambient light. Notably, plexiform neurofibromas have the potential for malignant transformation. No adverse reactions were reported; however, phototoxic reactions upon sun exposure, including severe pain and allergies, have been described in the literature, which may limit further implementation [75].

#### 5.1.6. Otolaryngologic Malignancies

Richard et al. reported on the feasibility of ICG NIR imaging for pediatric head and neck cancers, using an intravenous dosage of 1.5 mg/kg the day before surgery [43]. ICG NIR imaging revealed tumor extensions that were both undetectable on preoperative imaging and intraoperatively under ambient light and/or by palpation, facilitating more precise and safer resections. ICG NIR was also evaluated for local metastases and altered the surgical strategy in one case, prompting conversion to bilateral neck dissections. Despite ICG’s high sensitivity and specificity, positive tumor margins were observed in two of eight cases (25%), highlighting the need to combine this technique with other intraoperative imaging modalities, such as magnetic resonance imaging or ultrasound, to minimize residual disease.

In conclusion, while fluorescence-guided surgery with non-specific agents such as ICG appears promising for several pediatric solid tumors, its clinical utility remains unproven, with robust evidence currently only available for hepatoblastoma. For other tumor types, results are still exploratory and hampered by non-specific binding and variability in tumor histology and biology. Consequently, non-specific agents are prone to false-positive and false-negative signals, especially in highly vascularized or inflamed tissues. In tumors with infiltrative growth into surrounding normal tissue, FGS may therefore provide limited additional value for complete resection or tumor margins and may mislead surgical decision making. In addition, it may support intraoperative decision making that surgery might not be curative and/or that adjuvant treatment strategies may be more appropriate. To better define its value, a large prospective clinical trial (NCT04084067) has recently been initiated in the United States, including patients with osteosarcoma, Ewing sarcoma, rhabdomyosarcoma, non-rhabdomyosarcoma, neuroblastoma, renal and liver tumors, as well as other rare entities. This study will not only assess the added value of ICG-guided margin delineation and metastasis detection but will also explore factors such as uptake differences between primary and metastatic lesions, the influence of prior treatments, and the detection of residual disease.

### 5.2. Tissue-Specific Tumor Imaging

To address limitations in sensitivity and specificity associated with non-targeted agents, there is growing interest in targeted fluorescent agents. Several studies have highlighted the potential of targeted FGS for future applications in pediatric oncology [76,77]. However, progress in developing and implementing these agents remains limited, not only due to the scarcity of validated pediatric-specific targets, but also because of the limited availability of targeted agents, such as antibodies, that can be safely administered in children and that demonstrate the required high specificity for tumor cells with minimal binding to healthy tissues. Ideal imaging agents must therefore combine high tumor cell specificity and binding affinity with minimal off-target uptake and low toxicity. In adult cancers, antibodies and peptides directed against various oncological molecular targets such as the endothelial growth factor receptor, vascular endothelial growth factor A, carcinoembryonic antigen have been labeled with NIR dyes to delineate tumors, but such targets are not suitable in the pediatric setting due to the different origin of pediatric tumors [78,79]. Within pediatric oncology, two promising targets for FGS have recently emerged: the disialoganglioside (GD2) antigen, targeted by anti-GD2-based agents, for neuroblastoma and some sarcomas and the folate receptor, targeted by pafolacianine, for pulmonary metastases of (osteo)sarcoma. Until now, these agents have not yet been investigated in children.

#### 5.2.1. Targeting Neuroblastoma with Anti-GD2 Based Imaging Probes

Neuroblastoma surgery poses a high risk of complications as the tumor’s infiltrative growth into surrounding organs and vessels makes the resection challenging. While neoadjuvant therapy often induces tumor volume reduction, it also creates tissue heterogeneity, such as necrosis, fibrosis, matured neuroblastic tissue, and calcifications, which further complicates the distinction between tumor and healthy tissue [80]. Targeted FGS aims to allow for precise resection (≥95% tumor removal) of complex tumors while sparing vital adjacent structures, which both affect the prognosis and quality of life of the patient [81,82,83]. In neuroblastoma, this approach is particularly promising given the tumor-specific overexpression of GD2, which is minimally present in normal tissues and therefore represents an attractive target for both imaging and therapy. Anti-GD2, or Dinutuximab-beta, is currently used for immunotherapy in high-risk patients [84]. Importantly, GD2 is present on the various subtypes and stages of neuroblastoma and is retained on the cell membrane after induction chemotherapy, supporting its potential for targeted FGS [85].

Wellens et al. conjugated anti-GD2 to IRDye800CW, which showed specific binding to human neuroblastoma cells both in vitro and in vivo using xenograft mouse models [44]. Across patient-derived neuroblastoma organoids a universal, yet heterogeneous expression of GD2 was observed. Even tumors with low GD2 expression levels or tumors pretreated with anti-GD2 still provided real-time fluorescence signal. Similarly, Privitera et al. conjugated anti-GD2 to two NIR dyes, IRDye800CW and to IR12, and tested them in multiple cell lines [45]. Both conjugates are promising FGS probes and can be used in the NIR and shortwave infrared imaging (SWIR) window, with NIR providing more signal but SWIR providing sharper imaging at depth which could be beneficial for small lesions or lesions adhering to vital structures. To overcome limited depth penetration, Rosenblum et al. evaluated the potential of dual-labeled ^111^In-αGD2-IR800, consisting of both a radio-guided surgery (RGS) component and a FGS component in xenograft mouse models [46]. The gamma decay from the RGS component, ^111^In, can be used for finding the tumor localization at depth while the FGS component, αGD2-IR800, can be used for precisely visualizing the tumor margins for optimal resection. All studies demonstrated highly specific binding of anti-GD2 to neuroblastoma and promising results to use this antibody for FGS. The highest tumor to background (TBR) was achieved between two and six days [44,45,46]. To investigate anti-GD2-800CW in a clinical trial, a first-in-human phase Ib/II clinical trial has recently been opened in the Netherlands (EUCT 2023-507596-22-00).

#### 5.2.2. Targeting Pulmonary Metastases of (Osteo)Sarcoma with Pafolacianine

Surgery for pulmonary metastases is in some pediatric cancers essential, especially for tumors that are less responsive or resistant to adjuvant therapy, such as osteosarcomas and non-rhabdomyosarcomas soft tissue sarcomas [86]. Complete resection of these lesions is necessary to increase the chance of survival [87]. Similarly to ICG, targeted FGS could enhance surgical precision which could potentially influence surgical planning and long-term outcomes.

ICG has been widely applied to localize metastatic nodules. However, low specificity was observed and a need for more accurate techniques is preferred. Pafolacianine, also known as CYTALUX or OTL-38, targets the folate receptor and has successfully been used to improve surgical outcomes for primary and metastatic ovarian and lung tumors in adults [88,89]. In children, the folate receptor is also expressed in osteosarcomas [90]. Lehane et al. investigated the potential of using pafolacianine in four young adults (18–25 years) with pulmonary nodules of patients with osteosarcoma and Ewing sarcoma [47]. One additional lesion was found with the use of fluorescence in a patient with osteosarcoma, which was confirmed to contain malignant tissue. All osteosarcoma lesions that were identified under white light were fluorescent, of which two contained benign intraparenchymal lymph nodes. These false positive results were likely due to folate receptor-beta found on activated macrophages within benign lymph nodes. In the Ewing sarcoma patient, a 4 mm nodule was intraoperatively not distinguishable by fluorescence with faint fluorescence on back-table imaging, which was histologically proven malignant. These results showed that pafolacianine was safe and feasible but highlighted the need for careful target selection and validation in pediatric populations, where tumor biology may differ significantly from adults. Currently, two clinical trials started to investigate the potential of pafolacianine for pediatric pulmonary metastatic disease (NCT06235125) and for pediatric extracranial solid tumors (NCT06915727).

Collectively, these studies suggest that targeted FGS holds considerable promise in pediatric setting, especially for reducing the false positive results; however, the field is still waiting for the results of the first clinical trials.

## 6. Future Directions

Current evidence demonstrates that FGS in pediatric oncology is feasible and provides added value in selected applications. ICG has a well-established safety profile and has been shown to aid in lymph node mapping, hepatoblastoma surgery, and visualization of vascular structures and tissue perfusion, highlighting its key strengths. In these applications, FGS is most robust as it used to clarify anatomical details and less likely to mislead the surgical decision making. However, its non-specific nature, the predominantly anecdotal nature of the available evidence and the current lack of histopathological validation may limit broader use within pediatric oncology. In particular, ICG can visualize tumor tissue but does not specifically identify tumor lesions or define resection margins, residual viable tumor, or tumor-positive lymph nodes. These potential false-positives or false-negatives for malignancy could mislead surgical decision making and have a negative effect on the clinical outcome. Similar to adult oncology, the EPR effect is variable due to multiple factors including tumor type, volume, location, and the presence of necrosis and/or vascular mediators such as bradykinin and prostaglandins [73,91]. For example, inflammatory tissue express similarities to cancer in vascular mediator activity that can lead to false-positive fluorescence [92]. Increased fluorescence intensity in healthy tissue can also occur due to high vascularity [93]. Such strongly perfused tissues are prone to false-positive fluorescence signals, complicating intraoperative interpretation. Careful reporting of both tumor signal and surrounding tissue characteristics is therefore essential for optimizing the use of ICG, especially when looking at dosing and timing. These weaknesses also underscore the need for more specific fluorescence-guided approaches. Beyond improving intraoperative visualization, advances in FGS may also enable therapeutic extensions, such as photodynamic therapy. By combining tumor-specific fluorescence with light-induced cytotoxicity, these theranostic approaches could potentially enhance local tumor control while preserving surrounding healthy tissue [94,95]. However, in pediatric oncology, such applications remain largely exploratory and will critically depend on the development and validation of highly specific targeting agents tailored to the unique biological characteristics of pediatric tumors [95].

The development and validation of these targeted tissue-specific agents represents an important opportunity that may address these inaccuracies of non-specific agents improving sensitivity and specificity, ultimately enabling safer resections and more tailored treatment strategies. Current strategies for the development of tissue-specific agents often build on established targets from immunotherapy or adapt targets validated in adult oncology to pediatric cancers, as this may accelerate implementation compared to developing entirely novel targets. However, pediatric tumors frequently have an embryonal origin and differ substantially from adult malignancies in biological characteristics. Compared with adult cancers, they generally have a lower tumor mutational burden, which limits the availability of suitable molecular targets for tumor-specific imaging approaches, and they often arise in tissue types that are rarely encountered in adult oncology. Ideal targets should be highly expressed on tumor tissue with minimal off-target binding. Once identified, an appropriate targeting vehicle such as an antibody, antibody fragment, peptide, or small molecule must be selected, each with characteristic trade-offs in specificity, tissue penetration, and pharmacokinetics [96,97]. Antibodies offer high affinity but slower systemic clearance due to their larger size, which can result in prolonged background signal from circulating unbound agent and may require administration several days prior to surgery to allow sufficient clearance [98]. In contrast, smaller fragments and peptides provide faster tumor uptake and background clearance, sometimes at the expense of reduced stability. Small molecules are easier to synthesize and optimize, allow rapid clearance, but are generally less specific. The final step involves pairing the vehicle with a fluorescent dye compatible for intraoperative imaging.

Identified agents should undergo rigorous preclinical validation across in vitro, ex vivo, and in vivo models to confirm specificity, distribution, and safety prior to clinical translation [99]. Translation into humans is critical, as preclinical models cannot fully replicate pediatric tumor biology, and the safety profile of targeted agents must be carefully considered, which may pose a threat. Regulatory approval for first-in-child trials represents a key milestone, as fluorescent agents are generally classified as investigational medicinal products and are subject to drug development regulations. While comprehensive toxicology, pharmacokinetic, and manufacturing data are usually required, many standard tests for therapeutic medicinal products may be less relevant for optical agents, which are administered at much lower doses, that do not require any therapeutic effect. Consequently, early pediatric trials should focus on pharmacokinetics and safety at clinically relevant optical imaging doses, together with optimization of timing of administration. Placed in the broader landscape of emerging targeted optical agents, which promise greater molecular specificity but will demand even stricter methodological rigor, ICG offers a valuable benchmark. Its well-established safety profile and straightforward clinical implementation make it a practical first-line agent to determine which pediatric tumor types can be visualized with adequate sensitivity and specificity, and which will ultimately require the development of more tissue-specific next-generation agents.

To ensure reliable interpretation and broader translation, standardized and transparent reporting is essential, underscoring the need for more systematic and consistent quantitative assessment and reporting. This emphasizes the need for uniform definitions for quantitative metrics (tumor-to-background ratio, mean fluorescence intensity) to be able to compare clinical trials and different agents. These metrics are not only clinical variables, but do also reflects biological variability of tumors. As emphasized by Tummers et al., early-phase studies should incorporate: (1) intraoperative assessment at multiple timepoints and imaging settings to localize tumors, evaluate resection margins, and to identify distant lesions or lymph nodes; (2) specimen mapping with preserved orientation during bread-loafing and paraffin embedding to enable correlation between fluorescence and pathology; and (3) target validation in paraffin-embedded tissue by linking fluorescence to histology, thereby confirming binding specificity and assessing receptor heterogeneity [100]. In addition, quantitative parameters such as mean fluorescence intensity, sensitivity, and specificity must be consistently reported to allow comparison across studies and enable robust clinical translation.

Despite promising applications, FGS faces several inherent challenges. Interpretation remains highly dependent on the surgeon’s expertise, as differentiating between true and non-specific fluorescence can be subjective. Variability in fluorescence intensity, influenced by vascular inflow dynamics and tissue composition, further confuses intraoperative decision making. Additionally, tumor heterogeneity and complex tumor biology (e.g., coexistence of viable tumor, necrosis, and fibrosis) can affect the consistency and reliability of fluorescence signals. Nevertheless, histopathological confirmation remains indispensable to determine whether the observed fluorescence accurately represents malignant tissue and to validate the diagnostic performance of the technique. While quantitative methods may help standardize interpretation, no universally accepted protocols are currently available, limiting reproducibility and broader adoption.

## 7. Conclusions

In conclusion, the outlined steps represent a challenging yet clinically realistic and pediatric-specific roadmap for advancing fluorescence-guided surgery in pediatric oncology. Closing current gaps will require coordinated efforts in target discovery, imaging agent design, and clinical validation, as well as collaboration across pediatric oncology centers to standardize methods and reporting. If successful, fluorescence-guided surgery in pediatric oncology can evolve from a promising tool into an indispensable clinical approach that enhances surgical precision, reduces recurrence, and ultimately improves long-term outcomes for children with cancer.

## Figures and Tables

**Figure 1 cancers-18-00149-f001:**
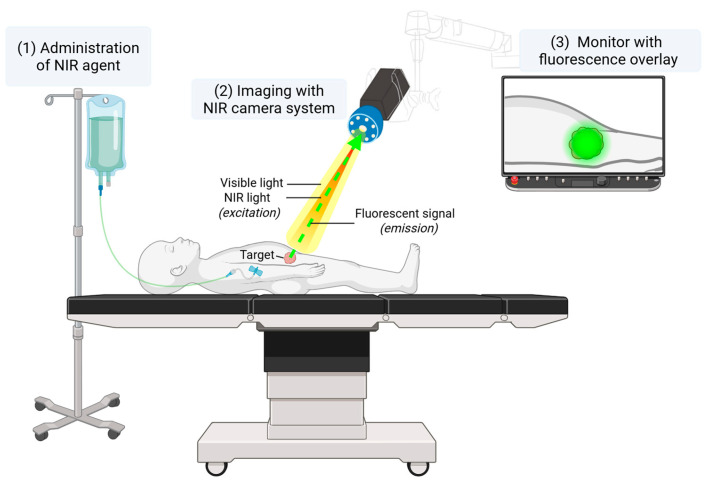
A schematic overview of a clinical FGS setup. A NIR fluorescent agent is administered intravenously or locally, either preoperatively or intraoperatively. The NIR camera system enables visible light as well as NIR light, which excites the fluorescent agent. The emitted light is then captured by the camera and displayed in real time on a monitor.

**Figure 2 cancers-18-00149-f002:**
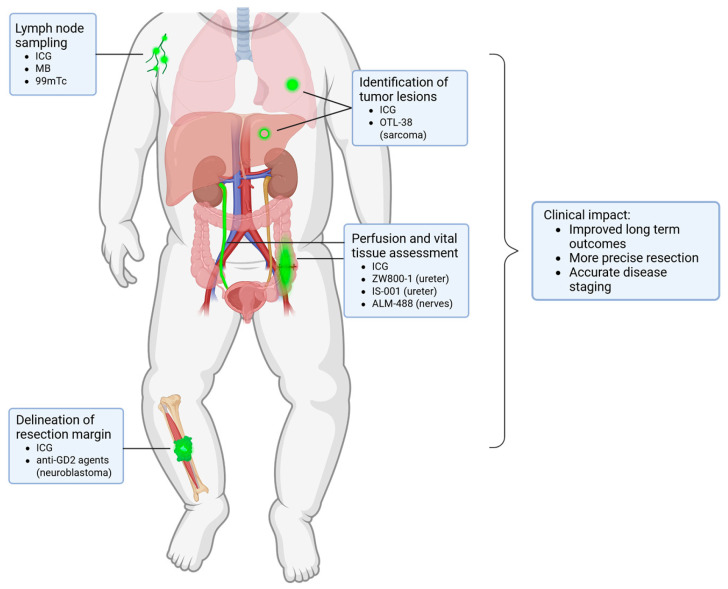
Overview of the clinical applications of FGS within pediatric oncology with the fluorescent agents per indication.

**Figure 3 cancers-18-00149-f003:**
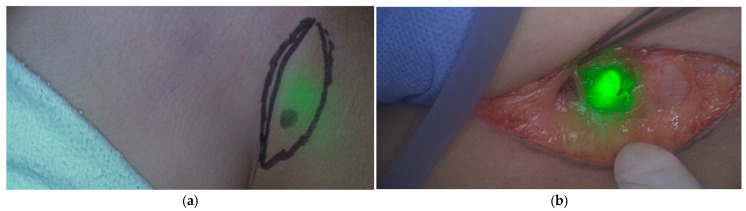
ICG fluorescence of the sentinel lymph node located in the inguinal region of a pediatric patient with melanoma. Pseudo-colored fluorescence overlay: (**a**) fluorescent lymph node observed through the skin before surgical incision; (**b**) fluorescent lymph node localized in vivo. Images were obtained from our Sentinel Lymph Node Procedure Study Using Fluorescence Imaging with ICG (NL71166.041.20).

**Figure 4 cancers-18-00149-f004:**
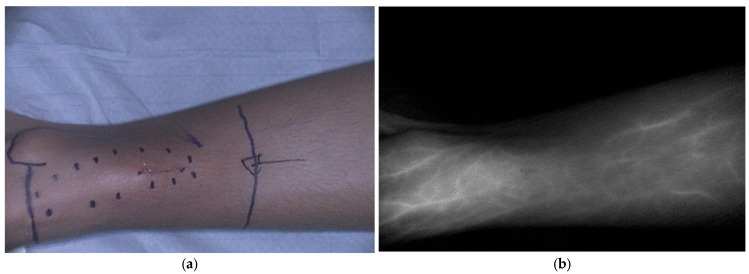
ICG fluorescence showing vascular perfusion, including the vena saphena magna, before primary bone tumor resection in a pediatric patient. (**a**) Bright field image; (**b**) Corresponding black-and-white fluorescence image. ICG was administered intravenously before surgery. Data derived from our clinical study Fluorescence with ICG in Sarcoma Surgery (PMC CRC 2024-003).

**Figure 5 cancers-18-00149-f005:**
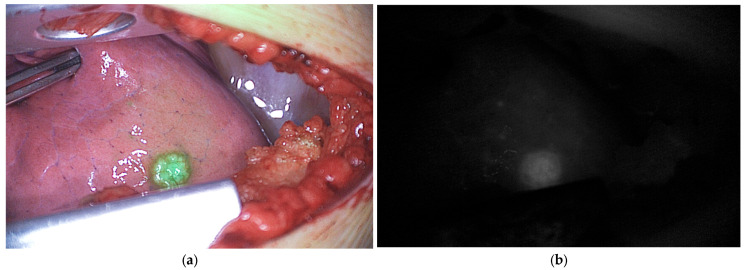
ICG fluorescence of vital pulmonary metastases in a pediatric patient with osteosarcoma during thoracotomy. (**a**) Pseudo-colored fluorescence overlay image; (**b**) Corresponding black-and-white fluorescence image. Data derived from our clinical study Fluorescence with ICG in Sarcoma Surgery (PMC CRC 2024-003).

**Table 1 cancers-18-00149-t001:** Overview of FGS in pediatric oncology.

Clinical Application	Clinical/ Preclinical	Population	Disease	Purpose	Imaging Agents	Citations
**Non-Specific Imaging**						
Lymphatic mapping, SNP	Clinical	Pediatric and young adults	Melanoma, myoepithelial neoplasm, squamous cell carcinoma and sarcoma	SNLB	ICG + 99 mTc or MB + 99 mTc 4 mg intraoperative ICG + 99 mTc + MB 1.25 mg intraoperative ICG + 99 mTc 0.25–5 mg intraoperative	[8,9,10,11]
	Clinical	Pediatric and (young) adults	Wilm’s tumor, (synovial) sarcoma, melanoma, squamous cell carcinoma, paratesticular rhabdomyosarcoma, renal tumors, myoepithelial neoplasm	Nodal sampling	ICG 5–10 mg intraoperative	[12,13,14,15]
Vital structures	Clinical	Pediatric	Sarcoma	Visualization of blood vessels and perisvascular system of nerves	ICG 0.1 mg/kg intraoperative	[16]
	Clinical	Pediatric	Abdominal masses and lymphoma	Vascular anatomy of mass, plane of resection during mesenteric division and perfusion assessment of bowel or organs	ICG 0.2–0.5 mg/kg intraoperative	[17,18,19,20,21]
Tumor imaging	Clinical	Pediatric	Hepatoblastoma	Identification of primary residual, and metastatic lesions	ICG 0.1–0.5 mg/kg, 24–96 h before surgery	[22,23,24,25,26,27,28,29,30,31,32,33]
	Clinical	Pediatric	Wilms’ tumor and renal cell carcinoma	Nephron-sparing surgery and identification of pulmonary (metastatic) lesions	ICG 1.5 mg/kg 24 h before surgery 2.5/5 mg/kg intraoperative	[34,35,36]
	Clinical	Pediatric	Abdominal tumors and lymphoma	Ovarian-sparing surgery, resection margins and vascularity of the abdominal mass	ICG 0.2–0.5 mg/kg intraoperative	[17,18,19,20,21]
	Clinical	Pediatric and young adults	Bone and soft tissue sarcoma	Guiding resection margins, identification of residual, and metastatic lesions	ICG 0.5–2.5 mg/kg 24 h before surgery	[36,37,38,39,40,41]
	Clinical	Young adults	Peripheral nerve sheath tumors	Localizing residual tumor tissue	Fluorescein 1 mg/kg after intubation	[42]
	Clinical	Pediatric	Otolaryngologic malignancies	Tumor extension	ICG 1.5 mg/kg 24 h before surgery	[43]
**Tissue-specific** **imaging**						
Tumor imaging	Preclinical	Mice models	Neuroblastoma	Tumor (margin) identification	Anti-GD2-IRDye800CW * Anti-GD2-IR800 & anti-GD2-IR12 * DPTA-aGD2-IR800 *	[44,45,46]
Metastases imaging	Clinical	Young adults	Pulmonary metastases of osteosarcoma and Ewing sarcoma	Tumor identification	Pafolacianine (Cytalux, OTL-38) 0.025 mg/kg 3–8 h before surgery	[47]

SNP: Sentinel lymph node procedure; ICG: indocyanine green; MB: methylene blue. * Preclinical dose/timing study.

## Data Availability

The data presented in this study are available on request from the corresponding author. Some of the data are not publicly available due to confidentially and in accordance with the Dutch Personal Data Protection Act.

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
