# Peer review of "Fluorescence-Guided Surgery in Pediatric Oncology: Current Practice and Future Directions"

_cancers, 2025, doi:10.3390/cancers18010149_

Round 1
Reviewer 1 Report
Comments and Suggestions for Authors
The authors have presented a comprehensive review of the current status of fluorescence guided surgery in pediatric oncology surgeries.
It is a well prepared review, showing the potentials and challenges in the field.
One minor suggestion: The authors may elaborate on the side effects or difficulty labelling of tumours or nerve with presumably labeled monoclonal antibodies in vivo.
Another area of contention is the additional benefit of, say, labelling ureter vs inserting ureteric catheters in safeguarding them.
Reviewer 2 Report
Comments and Suggestions for Authors
The manuscript presents that fluorescence-guided surgery shows promise for improving tumour removal in children, but its adoption is limited by non-specific dyes and the need for better clinical validation. Emerging targeted agents may help make it a reliable tool in pediatric oncology.
1) Restrict keywords to 5-6 2) The introduction should be comprehensively discussed, including fluorescence-guided surgery, image processing, and tumour-specific. 3) Please provide the schematic of the proposed work 4) Add recent relevant references to support the work: doi.org/10.1117/1.JBO.30.S1.S13709; doi.org/10.1016/j.yjpso.2023.100106 5) Figure 1 has unclear labelling and needs to be upgraded with a real-time scenario as a schematic representation. 6) Provide the SWOT analysis7) Discuss the limitations and challenges involved in the proposed work.
8) Highlight the novelty and significance
9) Add the comparison table with key parameters
10) Add more recent relevant references
11) The English language needs proper correction and continuation flow.
Reviewer 3 Report
Comments and Suggestions for Authors
This is a challenging topic to review as the concept of fluorescence guided surgery in paediatric oncology is at an early stage of development.
This is a comprehensive, well written, review of the topic and my comments are relatively minor. As the title is fluorescence guidance in paediatric oncology, there should be more discussion on the differences and similarities between adults and children. Clearly, the available data sets for children are limited, sometimes with not much more than anecdotal evidence.
Various examples of targeted and non-targeted agents are mentioned, but in most cases, it is not clear how useful the results are. Agents that help in clarifying anatomical details such as lymph nodes, ureters, nerves and blood vessels are clearly very useful, but agents that show false positives and false negatives for malignancy can mislead surgeons. When identifying multiple small metastases in organs like the liver and lungs the best value is probably in telling surgeons that the prospect of excising all of them is close to zero and that systemic treatments such as chemotherapy or field radiotherapy may be a better, or at least an adjuvant, option. An approach that combines fluorescence and treatment, such as photodynamic therapy (PDT) with the photosensitiser ALA, could be even better as PDT effects in normal tissues usually heal very well.
If fluorescence guidance gives precise delineation of a tumour, then clearly this makes surgery easier, but in many cases the value is limited, especially with cancers invading adjacent normal tissue and the risk of false positives. More should be said in the text about the actual value of these guidance options in the current state of the art. These problems are all described, but in many cases one is left not really knowing what the current state of the art is.
The description of fluorescence from tumours in individual organs is good, but it would be of value to add a column to table 1 showing the current status of FGS and the most important aspects with regard to future developments for each organ.
The general discussion on possible future developments is good.
Round 2
Reviewer 2 Report
Comments and Suggestions for Authors
The authors have made significant changes to the revised version, and it looks good; it can be accepted for publication.